# Modeling Between-Subject Variability in Subcutaneous Absorption of a Fast-Acting Insulin Analogue by a Nonlinear Mixed Effects Approach

**DOI:** 10.3390/metabo11040235

**Published:** 2021-04-12

**Authors:** Edoardo Faggionato, Michele Schiavon, Chiara Dalla Man

**Affiliations:** Department of Information Engineering, University of Padova, I35131 Padova, Italy; faggionato@dei.unipd.it (E.F.); michele.schiavon@dei.unipd.it (M.S.)

**Keywords:** biological variability, population modeling, insulin therapy, diabetes

## Abstract

Despite the great progress made in insulin preparation and titration, many patients with diabetes are still experiencing dangerous fluctuations in their blood glucose levels. This is mainly due to the large between- and within-subject variability, which considerably hampers insulin therapy, leading to defective dosing and timing of the administration process. In this work, we present a nonlinear mixed effects model describing the between-subject variability observed in the subcutaneous absorption of fast-acting insulin. A set of 14 different models was identified on a large and frequently-sampled database of lispro pharmacokinetic data, collected from 116 subjects with type 1 diabetes. The tested models were compared, and the best one was selected on the basis of the ability to fit the data, the precision of the estimated parameters, and parsimony criteria. The selected model was able to accurately describe the typical trend of plasma insulin kinetics, as well as the between-subject variability present in the absorption process, which was found to be related to the subject’s body mass index. The model provided a deeper understanding of the insulin absorption process and can be incorporated into simulation platforms to test and develop new open- and closed-loop treatment strategies, allowing a step forward toward personalized insulin therapy.

## 1. Introduction

The biological variability of insulin absorption and insulin action is the main obstacle to the optimal management of insulin treatment in type 1 (T1D) and type 2 diabetes (T2D). In fact, subcutaneous injections of the same exact insulin dose may greatly differ between different individuals or even in the same individual on different occasions [1]. From the masterwork of Binder in 1969 [2], several other papers were published aiming to investigate the reasons behind this variability. They overall agreed that the factors shaping the insulin concentration profile in blood are all those that influence insulin diffusion in the tissues, including the concentration, volume, and associated state (hexameric, dimeric, or monomeric) of injected insulin, the site and depth of injection, tissue blood flow, and skin temperature [3,4,5,6,7], but also, factors like the level of antibody binding seem to play a major role [8]. The considerable between-subject variability (BSV) and within-subject variability (WSV) complicate the tuning of the treatment performed by healthcare professionals, leading to dangerous glucose excursions, which are the main causes of acute and long-term diabetes complications.

Modeling the BSV and WSV of the subcutaneous kinetics of fast-acting insulin analogues represents an important step in understanding this phenomenon, which may help in better simulating the complex behavior of the glucose regulatory system and, in turn, improving both T1D and T2D management. The estimation of the variability present in a biological process is conventionally performed by the so-called “standard two-stage” method, which consists, first, of estimating the model parameters in each subject of the population and, then, assessing the mean vector and the covariance matrix [9], the latter being a quantitative measurement of the population variability. However, by doing so, the variability is usually overestimated. Moreover, the population estimates are not used to improve the individual ones, thus neglecting that a biological process shows a similar behavior in all the subjects who underwent the same experiment. Then, in the posterior analysis, one usually tries to detect some independent subject characteristics, normally referred to as covariates, that are, to some extent, related to the variability present in the process. However, this analysis does not employ the precision of the individual parameters in estimating the overall variability [10] and provides only a hint at the covariates to take into account, without investigating their actual contribution in the biological process [11].

A better approach to overcome these issues is nonlinear mixed effects (NLME) modeling. In this framework, subjects are assumed to be random realizations of a population whose distribution is shaped around some parameters (the so-called fixed effects). These population parameters are then employed to support the estimation of individual parameters, which are described using the deviation of each subject from the population value (random effects). All the available information is thereby employed in the estimation process. In addition to that, the estimation process can take advantage of prior information employing a Bayesian approach, which is known to be advantageous, especially in the case of a relatively small number of samples. Moreover, the NLME approach allows improving the model predictive power by introducing possible covariates in order to explain the portion of variability that is directly related to some macroscopic subject characteristic. In this way, the effective contribution of covariates to the population variability is estimated along with all the other model parameters. An final advantage comes from the possibility to use data coming from different studies even in the case of missing information or slightly different adopted protocols. By doing so, the features detected in the model are more robust since the influence of the setup of each study is reduced.

Recently, a model of subcutaneous absorption of fast-acting insulin analogues was developed and validated on a large database using standard modeling techniques [12,13]. Here, we aimed to apply the NLME modeling technique to such a model in order to describe the BSV present in the pharmacokinetics of subcutaneous absorption of lispro, a fast-acting insulin analogue. In doing so, we tried also to personalize the model by the use of subject covariates. Such a model will become an important component of the University of Virginia (UVa)/Padova T1D Simulator [14], an in silico platform accepted by the Food and Drug Administration (FDA) as a substitute for preclinical trials for certain insulin treatments, including closed-loop algorithms for artificial pancreas [15], recently used as an ideal test bench for the development and evaluation of glucose sensors [16] and novel insulin analogues [17,18]. Furthermore, the presented model could be incorporated also in the recently proposed Padova T2D simulator [19], allowing optimizing insulin therapy in this population. The incorporation into the simulators of a model that explicitly accounts for BSV would decrease the gap between the simulation environment and the real subject behavior, improving the in silico optimization of personalized insulin treatment. Finally, a model like this could also be employed in the development of model-based algorithms for subcutaneous insulin dosing in patients with T1D, like the model predictive controllers developed by the University of Cambridge [20] and by the collaboration between the Universities of Pavia, Padova, and Virginia [21].

## 2. Materials and Methods

### 2.1. Database and Protocols

We used data of 116 subjects with T1D from 3 clinical studies (Figure 1). All the subjects underwent a single subcutaneous injection of a fast-acting insulin analogue (lispro). The database contained also information about the demographic (age) and anthropometric (body weight (BW), body height (BH), body mass index (BMI), and body surface area (BSA)) features of the subjects. All the observed measurements below the quantification limit (BQL) were discarded from the analysis, together with very unreliable measurements due to issues with insulin assays. All studies were performed at the Profil Institute for Metabolic Research in Neuss, Germany, in accordance with the principles of the Declaration of Helsinki and the local Ethics Committee. All patients signed an informed consent prior to study start. The three studies are briefly described below, and more detailed information about the database and the adopted protocols are available in [12].

#### 2.1.1. Study 1

In the first study [22] (Figure 1, Panel A), forty-two T1D subjects (age = 42±11 years, body weight = 74±10 kg, height = 175±8 cm) underwent a euglycemic clamp and received a subcutaneous injection of 12 U of insulin lispro at around noon (t=0), after an overnight fast. During the clamp, glucose infusions were used to keep blood glucose stable at around 100 mg/dL and avoid the patient entering a hypoglycemic state during the experiment. The blood samples were collected at t= 0, 3, 7, 10, 15, 20, 25, 30, 45, 60, 90, 120, 150, 180, 210, 240, 300, 360, 420, and 480 min to measure glucose and insulin levels. A specific radio immunoassay was used for the determination of insulin lispro levels (BQL equal to 5 μU/mL).

#### 2.1.2. Study 2

In the second study [12] (Figure 1, Panel B), thirty-seven T1D subjects (age = 47±12 years, body weight = 80±8 kg, height = 180±5 cm) underwent a euglycemic clamp and received a subcutaneous injection of 0.2 U/kg of insulin lispro at around noon (t=0), after an overnight fast. During the clamp, glucose infusions were used to keep blood glucose stable at around 100 mg/dL and avoid the patient entering a hypoglycemic state during the experiment. The blood samples were collected at t= 0, 4, 8, 12, 16, 20, 25, 30, 35, 40, 45, 50, 55, 60, 70, 80, 90, 105, 120, 135, 150, 180, 210, 240, 300, 360, 420, 480, 540, 600, 660, and 720 min to measure glucose and insulin levels. A radio immunoassay was used for the determination of insulin lispro levels (BQL equal to 5.5
μU/mL).

#### 2.1.3. Study 3

In the third study [12] (Figure 1, Panel C), thirty-seven T1D subjects (age = 44±13 years, body weight = 81±10 kg, height = 180±8 cm) consumed a standardized meal and received a subcutaneous injection of 0.2 U/kg of insulin lispro at around noon (t=0), after an overnight fast. The meal was served to maintain high blood glucose level and avoid the subject entering a hypoglycemic state during the experiment. The blood samples were collected at t= 0, 4, 8, 12, 16, 20, 25, 30, 35, 40, 45, 50, 55, 60, 70, 80, 90, 105, 120, 135, 150, 180, 210, 240, 270, 300, 360, 420, and 480 min to measure glucose and insulin levels. A radio immunoassay was used for the determination of insulin lispro levels (BQL equal to 5.6
μU/mL).

### 2.2. The Nonlinear Mixed Effects Model

In this work, we used an NLME model to describe the BSV of subcutaneous absorption of a fast-acting insulin analogue. A fundamental ingredient of this approach is the availability of a large database preferably including also subject’s demographic and anthropometric data, known as covariates, which could be used to assess the influence, if any, of such factors on individual data (Figure 2, left panels). The NLME model here developed consisted of two sub-models: a structural model (the so-called intra-individual model, here composed by a set of differential equations, Equation (Equation 3)) describing the physiological process of interest and a stochastic model (the so-called inter-individual model, Equations (Equation 4) and (Equation 5)) describing the between-subject variability in the physiological process of interest, eventually including some covariates (Figure 2, central panel). Finally, the unexplained source of variance was assumed to be the measurement error noise, which was also described by a proper error model (Equation (Equation 7)). The general form of an NLME model is given by the following set of equations:(1)$$z_{j}=f_{j}\left(x_{j},\psi_{j}\right)+v_{j}$$
(2)$$\psi_{j}=d\left(c_{j},\theta ,\eta_{j}\right)$$
where: $z_{j}$ is the measurement vector for the *j*th subject, $x_{j}$ is a vector incorporating variables such as time and injected dose, $f_{j}$ is a vector function describing the relation among $z_{j}$, $x_{j}$, and $\psi_{j}$, the set of model parameters, while $v_{j}$ is the intra-individual error vector. In addition, $d_{j}$ is a ψ-dimensional vector function describing the relation between $\psi_{j}$ and $c_{j}$, the covariates vector, θ, the population parameters (fixed effects), and $\eta_{j}$, the inter-individual error (random effects). Once having defined its structure, the NLME model was identified using the insulin concentration data of the three datasets, together with subject covariates. The identification process provided an estimate of the fixed effects ($\hat{\theta }$) and of the variability of the random effects in the population ($\hat{\Omega }$), as well as the individual parameter estimates ($\hat{\psi_{j}}$). These estimates were finally used to obtain the data fits (Figure 2, right panels).

#### 2.2.1. Physiological Model of Insulin Kinetics

Here, the physiological model of insulin kinetics (Equation (Equation 3)) was the linear three-compartment model (Figure 3) recently proposed by Schiavon et al. [12] for insulin lispro. The first two compartments, $I_{sc1}$ and $I_{sc2}$, represent insulin masses in the subcutis, in a non-monomeric and monomeric state, respectively. From the former, insulin can either be absorbed in the plasma insulin compartment $I_{p}$ with a rate constant $k_{a1}$ (min${}^{-1}$) or decomposed into monomers with a rate constant $k_{d}$ (min${}^{-1}$). From the latter, insulin can only be absorbed in the plasma insulin compartment $I_{p}$ with a rate constant $k_{a2}$ (min${}^{-1}$). Finally, insulin is degraded, usually by liver and kidneys, and this process is represented in the model by the rate constant $k_{e}$ (min${}^{-1}$). The dose is injected in the $I_{sc1}$ compartment, while measurements are collected from the plasma compartment $I_{p}$, which has a distribution volume of $V_{I}$ (L/kg). The model accounts also for a subject-specific delay in the insulin absorption through the parameter τ (min). The equations of the model are:(3)$$\left\{\begin{array}{c} {\dot{I}}_{sc1}\left(t\right)=-\left(k_{a1}+k_{d}\right)I_{sc1}\left(t\right)+u\left(t-\tau \right)\\ {\dot{I}}_{sc2}\left(t\right)=-k_{a2}I_{sc2}\left(t\right)+k_{d}I_{sc1}\left(t\right)\\ {\dot{I}}_{p}\left(t\right)=-k_{e}I_{p}+k_{a1}\left(t\right)I_{sc1}\left(t\right)+k_{a2}I_{sc2}\left(t\right)\\ y\left(t\right)=I_{p}\left(t\right)/V_{I} \end{array}\right.$$

#### 2.2.2. Model of the Parameter Variability

The model of the parameter variability (Equation (Equation 2)) is represented by a set of equations describing the BSV of the physiological parameters (Equations (Equation 4) and (Equation 5)). In this work, several models of increasing complexity were tested and compared. They all share the following basic assumptions:(4)$$\psi_{j}=\theta \;\exp\left(\eta_{j}\right)$$
(5)$$\eta_{j}∼N\left(0,\Omega \right)$$
i.e., the distribution of the random effects $\eta_{j}$ is assumed to be Gaussian, so that the physiological parameters belong to a log-normal distribution. In addition, the stochastic model can include some covariates of the subjects to explain the dependency of the parameter on the subject characteristics:(6)$$\psi_{i,j}=\theta_{i}\;\exp\left(\beta_{k,i}\;\left(c_{k,j}-\bar{c_{k}}\right)+\eta_{i,j}\right)$$
where $c_{k,j}$ is the *k*th covariate of the *j*th subject and $\bar{c_{k}}$ is the reference value for that covariate (here set to the mean value of our database). For each covariate added to the model, a coefficient $\beta_{k,i}$ has to be estimated. Using the deviation from a reference value instead of the raw value of the covariate is a common strategy [11], which brings two main advantages: The first advantage is that if there is any subject with a missing covariate value, the reference value can be attributed to them, so that this subject does not influence the estimation of the associated coefficient $\beta_{k,i}$, and their data can still be used to estimate the population parameters; however, this procedure could bias the estimates in the case of missing values. Anyway, this was not the case here, since all the information was available from the database. The second advantage is that the parameter $\beta_{k,i}$ can be interpreted more easily as the contribution made to the model parameter $\psi_{i}$ by a unitary deviation of $c_{k}$ from the reference value. Finally, it is possible to introduce some correlations between the model parameters by appropriately defining the covariance matrix Ω of the random effects η. For each correlation introduced in the covariance matrix, a parameter $\rho_{i_{1},i_{2}}$ has to be estimated. The variability models to test were chosen applying a Pearson’s correlation test to the estimated parameters in order to detect the most correlated covariate-parameter pairs and random effects. In this work, a set of 14 variability models incorporating different combinations of correlations between the random effects and covariates was considered.

#### 2.2.3. Measurement Error Model

The model for the measurement error is:(7)$$v\left(t\right)=\sqrt{a^{2}+b^{2}\;y{\left(t\right)}^{2}}\;\;\epsilon \left(t\right)$$
where the variable y(t) is the output of the physiological model, i.e., the plasma insulin concentration, and ϵ(t) is a Gaussian random variable, i.e., ϵ(t)∼N(0,1). For small values of y(t), v(t) presents an almost constant standard deviation (*a*), while, for high values of y(t), v(t) has almost a constant coefficient of variation (*b*). Of note, since the three studies adopted different protocols and utilized different insulin assays, the error model was identified separately for each database. Therefore, a total of 6 error parameters were estimated: $a_{1}$ and $b_{1}$ for the first study, $a_{2}$ and $b_{2}$ for the second, and $a_{3}$ and $b_{3}$ for the third.

### 2.3. Parameter Estimation

The physiological model is a priori non-uniquely identifiable, since the rate parameters $k_{d}$ and $k_{a2}$ are interchangeable. To overcome this issue, it was assumed $k_{d}\geq k_{a2}$, without loss of generality. To implement this inequality in the model, we defined $\alpha =k_{d}-k_{a2}$ and assumed that it belongs to a log-normal distribution, so that α>0. Therefore, the unknown parameters to be estimated are the 6 fixed effects of the physiological model (τ, $\;V_{I}$, α, $\;k_{a1}$, $\;k_{a2}$, and $\;k_{e}$), the 6 standard deviations of the random effects, describing the BSV of the physiological parameters ($\;\omega_{\tau }$, $\;\omega_{V_{I}}$, $\;\omega_{\alpha }$, $\;\omega_{k_{a1}}$, $\;\omega_{k_{a2}}$, and $\;\omega_{k_{e}}$), the 6 parameters related to the measurement error ($a_{1}$, $b_{1}$, $a_{2}$, $b_{2}$, $a_{3}$, and $b_{3}$), and possibly coefficient $\beta_{k,i}$ to account for covariates and $\rho_{i_{1},i_{2}}$ to account for correlations in the random effects. Hence, for each model, a minimum of 18 population parameters were identified.

For model identification and validation, we used the software Monolix (*Monolix* 2020R1 [23], ©Lixoft, Antony, France), which implements the stochastic approximation of expectation maximization (SAEM) in combination with a Markov chain Monte Carlo (MCMC) method to estimate the maximum likelihood of the NLME model parameters [24]. The fixed effects were initialized to the median of the parameter estimates found in [12] and the standard deviation of the random effects *w* to 0.1, while all the other parameters were left to their default initial values. As done in [12], a priori information on the volume of distribution $V_{I}$ [25] was added to improve the a posteriori identifiability of this parameter. The Fisher information matrix was estimated with the Metropolis–Hastings algorithm (an MCMC method) and the likelihood through an importance sampling method [23].

### 2.4. Model Assessment and Comparison

Model performance was assessed in terms of the residual distribution, the physiological plausibility of the parameters, the precision of the estimates, and the parsimony criteria. In particular, the goodness of the individual fits was checked by the visual inspection of the data vs. individual predictions. In addition, residual distribution provided by the individual weighted residuals (IWRES), defined as the difference between the measurement z(t) and the model prediction $\hat{y}\left(t\right)$ weighted by the standard deviation of the error v(t):(8)$$\mathrm{IWRES}=\frac{z\left(t\right)-\hat{y}\left(t\right)}{\sqrt{a^{2}+b^{2}\;\hat{y}{\left(t\right)}^{2}}}$$
was compared to a standard Gaussian distribution. Moreover, Monolix allows checking the normality of the IWRES with a Shapiro–Wilk (SW) test. However, the performed SW test is inadequate if the sample size is large (the software documentation itself claims this point [23]). To avoid this issue, the Shapiro–Wilk test was applied on the residuals of each subject to assess normality, while the runs test was used to assess randomness. This analysis was performed in R (*R* 4.0.3 [26], ©The R Foundation, Vienna, Austria) with a significance level p=0.05. The distributions of the kinetic parameters were then checked to be physiologically plausible. The precision of the estimates was obtained from the inverse of the Fisher information matrix $I_{F}\left(\hat{\theta }\right)$ and summarized by reporting the average relative standard error (RSE) and number of parameters with RSE > 50%. The RSE for the parameter $\theta_{i}$ is calculated as:(9)$$\mathrm{RSE}\left({\hat{\theta }}_{i}\right)=\frac{\sqrt{C_{i,i}\left(\hat{\theta }\right)}}{{\hat{\theta }}_{i}}$$
where $C_{i,i}\left(\hat{\theta }\right)$ is the *i*th diagonal element of the covariance matrix of the parameters:(10)$$C\left(\hat{\theta }\right)=J^{T}I_{F}{\left(\hat{\theta }\right)}^{-1}J$$
and the transformation using the Jacobian J is necessary since $I_{F}\left(\hat{\theta }\right)$ is obtained with the likelihood computed using the normally transformed parameters $\hat{\theta }$. Moreover, the Bayesian information criterion corrected for the NLME model (BICc) [27] was used to compare models that produced a satisfactory score in all the previous metrics:(11)$$\mathrm{BICc}=-2\ln\left(L_{z}\left(\hat{\theta }\right)\right)+\ln\left(N\right)P_{R}+\ln\left(n_{tot}\right)P_{F}$$
where $L_{z}\left(\hat{\theta }\right)$ is the likelihood reached by the NLME model on the measurement data z, *N* is the number of subjects, $P_{R}$ is the number of estimated parameters of the variability model (i.e., the standard deviations of the random effects ω, possible correlations ρ, and eventual covariates β), and $P_{F}$ is the number of fixed effects parameters of the physiological model (θ) plus the number of error coefficients (*a* and *b*), while $n_{tot}$ is the total number of measurements. Finally, correlation analysis among random effects, as well as random effects and subjects’ covariates (or a logarithmic transformation thereof) was performed to test if parameters $\rho_{i_{1},i_{2}}$ or $\beta_{k,i}$, respectively, deserved to be introduced into the model.

## 3. Results

### 3.1. Comparison of the Variability Models

As shown in Table 1, all the attempted models provided acceptable residuals, with at least 89.7% and 87.8% of the subjects that passed the Shapiro–Wilk test and the runs test, respectively. Moreover, the models provided also physiologically plausible estimates. Therefore, for the selection of the best model, we used the precision of the estimates and the parsimony criterion. Models 4, 5, 8, 12, and 14 were discarded since they provided either a too high mean RSE or at least one parameter with RSE > 50%. Among the remaining models, Model 11 was the one providing the lowest BICc, and therefore, it was selected as the most parsimonious one. Compared with the models reported in Table 1, this model provides the second best BICc, the third best average precision of the estimates, as well as independent residuals and physiologically plausible estimates of the parameters.

### 3.2. Selected Model of the Parameter Variability

Here, the formulas of the variability Model 11 are reported:(12)$$\psi :\left\{\begin{array}{c} \tau =\tau^{pop}\exp\left(\eta_{\tau }\right)\\ V_{I}=V_{I}^{pop}\exp\left(\eta_{V_{I}}\right)\\ \alpha =\alpha^{pop}\exp\left(\eta_{\alpha }\right)\\ k_{a1}=k_{a1}^{pop}\exp\left(\eta_{k_{a1}}\right)\\ k_{a2}=k_{a2}^{pop}\exp\left(\beta_{BMI,k_{a2}}\left(BMI-\bar{BMI}\right)+\eta_{k_{a2}}\right)\\ k_{e}=k_{e}^{pop}\exp\left(\eta_{k_{e}}\right) \end{array}\right.$$
with the following covariance matrix of the random effects η:(13)$$\Omega =\left[\begin{array}{cccccc} \omega_{\tau }^{2} & 0 & 0 & 0 & 0 & 0\\ 0 & \omega_{V_{I}}^{2} & 0 & 0 & \omega_{V_{I}}\omega_{k_{a2}}\rho_{V_{I},k_{a2}} & \omega_{V_{I}}\omega_{k_{e}}\rho_{V_{I},k_{e}}\\ 0 & 0 & \omega_{\alpha }^{2} & 0 & 0 & 0\\ 0 & 0 & 0 & \omega_{k_{a1}}^{2} & 0 & 0\\ 0 & \omega_{V_{I}}\omega_{k_{a2}}\rho_{V_{I},k_{a2}} & 0 & 0 & \omega_{k_{a2}}^{2} & \omega_{k_{a2}}\omega_{k_{e}}\rho_{k_{a2},k_{e}}\\ 0 & \omega_{V_{I}}\omega_{k_{e}}\rho_{V_{I},k_{e}} & 0 & 0 & \omega_{k_{a2}}\omega_{k_{e}}\rho_{k_{a2},k_{e}} & \omega_{k_{e}}^{2} \end{array}\right]$$

It is worth noting that all the physiological parameters were assumed non-negative and modeled as log-normal random variables. In addition, the model includes the BMI as a covariate in the equation of the parameter $k_{a2}$ through the parameter $\beta_{BMI,k_{a2}}$ and the correlation between the random effects of the parameters $V_{I}$, $k_{a2}$, and $k_{e}$ through the parameters $\rho_{V_{I},k_{a2}}$, $\rho_{k_{a2},k_{e}}$, and $\rho_{k_{a2},k_{e}}$, respectively.

### 3.3. Validation of the Selected Model

Model prediction of the plasma insulin concentration was satisfactory, as can be seen in the left-hand panels of Figure 4, where data vs. model prediction of one representative subject for each of the three studies is reported. Moreover, the distributions of the IWRES obtained for each study compared well against a standard Gaussian distribution (Figure 4, right panels). With the selected model, 91.4% of the subjects passed the Shapiro–Wilk test for normality, and 88.5% percent passed the runs test for randomness.

In Table 2, the estimates of the population parameters are reported together with their precision expressed as the RSE. The parameters were estimated with a mean RSE of 9.11%, and no parameter reported an RSE > 50%. The quantiles of the distributions of the estimated individual parameters are reported in Table 3. All the individual parameter estimates were within physiological plausible ranges and compared well against the fixed effects of the population parameters. This is also shown in Figure 5 where the distributions of the estimated individual parameters are reported: in the left panels, the estimates of the individual parameters of the physiological model are compared with their theoretical distribution (i.e., the log-normal distribution derived from Equation (Equation 12) by knowing the estimated fixed effects reported in Table 2), while in the right panels, the estimates of the random effects η are compared with a standard Gaussian distribution. A final overview of the model performance can be done using a visual predictive check (VPC, Figure 6). In this graph, the percentiles of data obtained from multiple MCMC simulations of the final model (displayed as 90% prediction intervals, blue and red bands) are compared with the percentiles of the observed data (black lines). The VPC shows that the model was able to reproduce both the average trend (50th percentile, red band), as well as the variability (10th and 90th percentiles, blue bands) of the observed data well. The presence of outliers (red circles) in the bottom part of the graph is most likely due to having discarded BQL data from the original dataset, having thus introduced a bias in the computation of the empirical percentiles, which were, therefore, slightly overestimated.

## 4. Discussion

Despite the progress made in insulin preparation, the timing and dosing of insulin therapy are still inadequate to achieve optimal glucose control in T1D, mainly due to the high variability in the absorption process. The dosing regimens are currently tuned by the physician by a trail-and-error approach, which could lead to dangerous fluctuations in the blood glucose levels of the patient. Hence, a model able to describe that variability would be a powerful tool to optimize and personalize insulin initiation and titration.

In this work, a reliable NLME model describing absorption of fast-acting insulin analogues from a subcutaneous injection was presented. This model was given by the combination of the compartmental kinetic model developed in [12] with a new model describing the parameter variability, which represents an important step forward toward the description of BSV of subcutaneous insulin absorption. The model was identified on a large database of 116 T1D subjects using the Monolix built-in SAEM and MCMC algorithms. The model was able to reproduce the measurement data well, with an IWRES distribution that followed a standard Gaussian distribution in all three studies. In support of this, ninety-one-point-four percent of subjects passed the Shapiro–Wilk test to assess the normality of the residuals, while 88.8% passed the runs test to assess the randomness. The 22 model parameters were estimated with good precision: the mean RSE was equal to 9.11%, and no RSE was greater than 50%. Furthermore, the distributions of the individual parameters were within the physiological ranges and close to the values reported in [12]. The final validation step using the VPC (Figure 6) showed how well the model was able to capture both the typical trend and the variability of the absorption process.

The final covariance matrix of the random effects Ω (Equation Equation 13) was suggested and then confirmed by the posterior correlation analysis performed on the estimated random effects. From the results of Model 1, the $\rho_{V_{I},k_{a2}}$ correlation was detected (correlation = 0.40, *p*-value < $10^{-5}$), then from Model 2, the $\rho_{V_{I},k_{e}}$ and $\rho_{k_{a2},k_{e}}$ correlations were detected (correlation = −0.30 and −0.27, respectively, both *p*-values < 0.01), and finally, from Model 3 and on, no significant correlation between the random effects was detected (all *p*-values > 0.05).

Conversely, the detection of covariates to be introduced into the model was not that straightforward. For instance, although Pearson’s correlation test strongly suggested the introduction of the covariate BMI in the variability model of $V_{I}$ (Model 5), this did not bring any additional information since $V_{I}$ is already expressed in liters per kilograms of BW, and the BMI directly derives from the BW, leading to poor precision in the estimation of $\beta_{BMI,V_{I}}$. The final model (Model 11) included the covariate BMI that directly explained a portion of the variance of the parameter $k_{a2}$, which is the fractional rate of absorption from the subcutaneous space to the circulation, in agreement with the posterior statistical analysis performed in [12]. In the NLME framework, it was then possible to estimate the contribution of a deviation from the BMI reference value to the parameter $k_{a2}$ through the parameter $\beta_{BMI,k_{a2}}$: for each unit of BMI, the logarithm of $k_{a2}$ decreased by a factor of −0.087. This negative relation between the BMI and $k_{a2}$ seems physiologically reasonable since the higher the BMI of the subjects, the thicker is their skin and adipose layer and therefore the higher will be the time taken for insulin to be absorbed through the subcutaneous tissue into plasma. All the attempts to further improve the final model failed, even if Model 14 was close to succeeding. This model was identical to Model 11 with the addition of the log-transformed age in the variability model of τ. However, the parameter $\beta_{\ln(AGE),\tau }$ was estimated with an RSE of 59.7%, and therefore, this model was discarded. Nevertheless, this could still indicate a hidden contribution of age in the delay of insulin absorption, which was not detected by our analysis. Actually, the parameter τ summarizes a complex kinetic, which is sometimes described with one or more compartments in the physiological model [12] and that may hide a parameter highly correlated with age. Moreover, the delay could also be influenced by the experimental setup, which differed among the three studies, and the relation could even be nonlinear, such as the skin thickness, which presents an inversion of the trend with age increasing [28].

In the literature, different error models for insulin assays have been proposed, for example: the standard deviation of the measurement error was assumed constant in [12], proportional to the observed insulin in [29], and a combination of a constant and a proportional factor in [30]. The combined error model that we selected (Equation (Equation 7)) was confirmed by the fact that the error coefficients *a* and *b* were not fixed, but estimated alongside all the other model parameters with good precision and by the IWRES distributions that followed a standard Gaussian curve (Figure 4), suggesting this error model for each of the three studies forming the database.

Possible limitations of this study were that the model was identified only on insulin lispro as representative of commercially available insulin compounds and that, in all the experiments, each subject underwent a single subcutaneous injection of lispro, while multiple daily injection therapy requires the injection of different types of insulin administered more times in a day. Moreover, the analysis of the WSV was not allowed since each subject was examined only once, as per the protocols. In addition, a relatively small number of covariates was analyzed: only three of them were independent (age, BW, and height), and only a log-transformation was tested along with the raw value. Furthermore, one can argue that the estimated variability could be an underestimation of the variability present in day-to-day life since, during inpatient studies, factors like physical activity, which can also influence insulin absorption [5], were absent. The above limitations may be overcome in future studies designed to further validate and improve the model, e.g., testing the model on other databases where different types of insulin analogues are utilized or where different protocols are adopted that may allow modeling the WSV of the process, thus providing a more robust tool for the optimization of insulin treatment. Finally, the same methodologies can be applied to describe the BSV present in the absorption of long-acting insulin analogues and insulin action in order to estimate the whole variability present in the glucose-insulin system, from insulin absorption to glucose response. All the analysis was performed using Monolix (proprietary software, free for academic users [23]) and a minor part using R (open-source software [26]); nevertheless, the analysis could be performed entirely with R or using other software, e.g., NONMEM (proprietary software, *NONMEM* [31], ©ICON, Dublin, Ireland).

## 5. Conclusions

In this work, an NLME model aiming to describe the BSV that affects subcutaneous absorption of fast-acting insulin analogues was proposed. The model was built by adding, to an already existing physiological model, a stochastic model describing the BSV of model parameters and their relationship with subject covariates. The model, selected among a collection of 14 models, was able to accurately predict insulin appearance in plasma from subcutaneous injection, as well as to provide an estimate of the BSV. This will help in better understanding the factors that influence the subcutaneous absorption of insulin analogues. Together with the mathematical modeling of the BSV, another step forward made by this work is the identification of a correlation between the BMI and the rate of insulin absorption from subcutaneous tissue into plasma. Actually, being able to account for subject variability and to detect useful covariates will pave the way for precision medicine in the field of diabetes, providing a customized approach to the disease for each patient. Furthermore, this model will be useful in the development of algorithms for subcutaneous insulin delivery implemented in insulin pump devices [20,21] and will be an important component of in silico platforms, like the UVa/Padova T1D Simulator [14,15,16,17,18,19]. The incorporation into the simulator of models that account for subject variability would allow for more realistic simulations, providing great benefits on the way to the development and approval of new insulin compounds. In this way, the dosing regimens could be optimized, improving the insulin therapy of millions of diabetic patients, in a cost-effective way.

## Figures and Tables

**Figure 1 metabolites-11-00235-f001:**
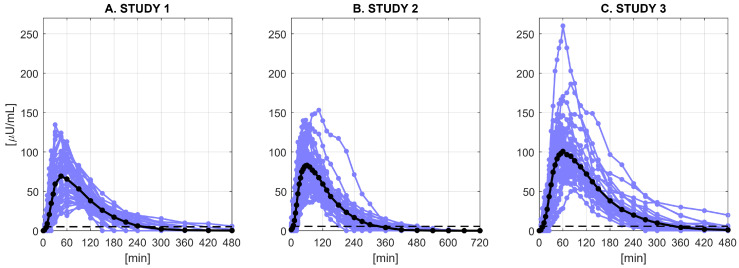
Plasma insulin concentration (μU/mL) in Study 1 (**A**), Study 2 (**B**), and Study 3 (**C**). Blue lines represent individual profiles, black lines the mean profile of each dataset, and dashed lines the BQL of each dataset.

**Figure 2 metabolites-11-00235-f002:**
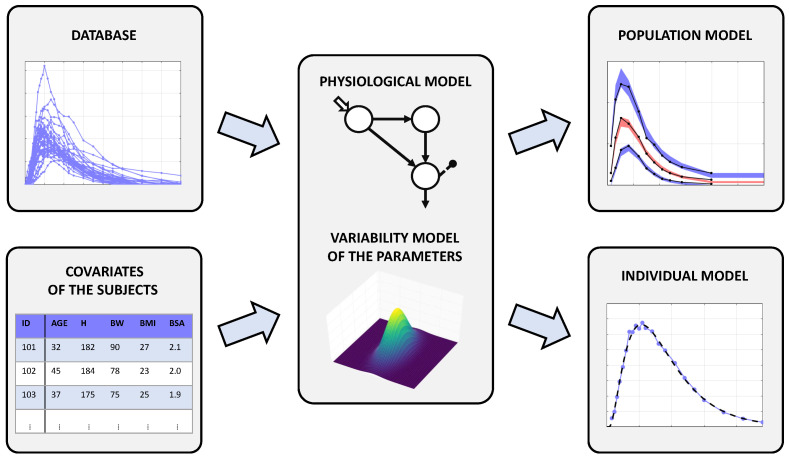
Schematic representation of the NLME approach. The central panel represents an NLME model: it consists of a structural model of the system coupled with a model of the parameter variability. Feeding the NLME model with a large database containing also subject covariates, one can obtain both the population and the individual models better fitting the data.

**Figure 3 metabolites-11-00235-f003:**
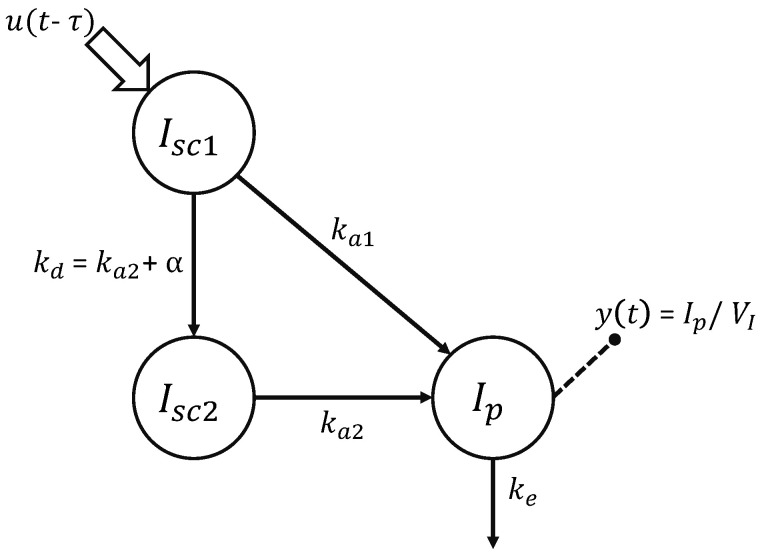
Compartmental representation of the model describing the subcutaneous absorption of fast-acting insulin [12]. Model compartments are represented with circles, and the fluxes are indicated with arrows. Signals *u* and *y* represent the subcutaneous insulin administration and plasma insulin concentration, respectively.

**Figure 4 metabolites-11-00235-f004:**
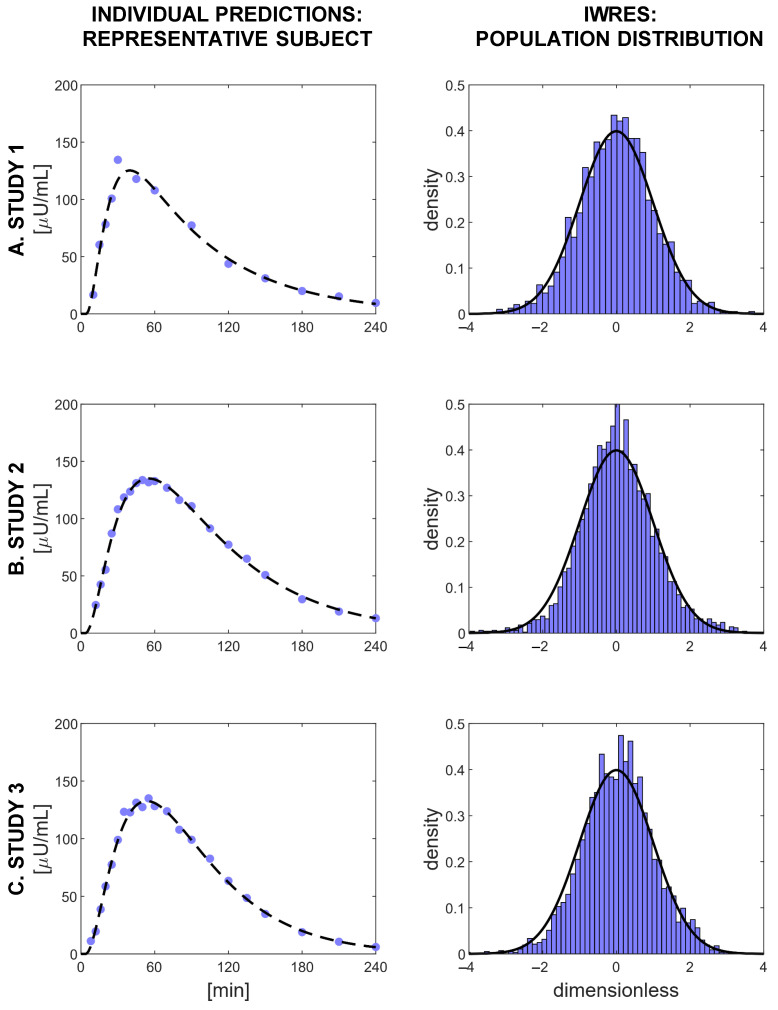
Evaluation of the goodness of the predictions of Model 11. Left panels: the fit of the model for three representative subjects, one for each study. Blue dots indicate plasma insulin measurements, and black dashed lines represent the model predictions. Right panels: the distributions of the IWRES in each study, compared with a standard Gaussian distribution (black line).

**Figure 5 metabolites-11-00235-f005:**
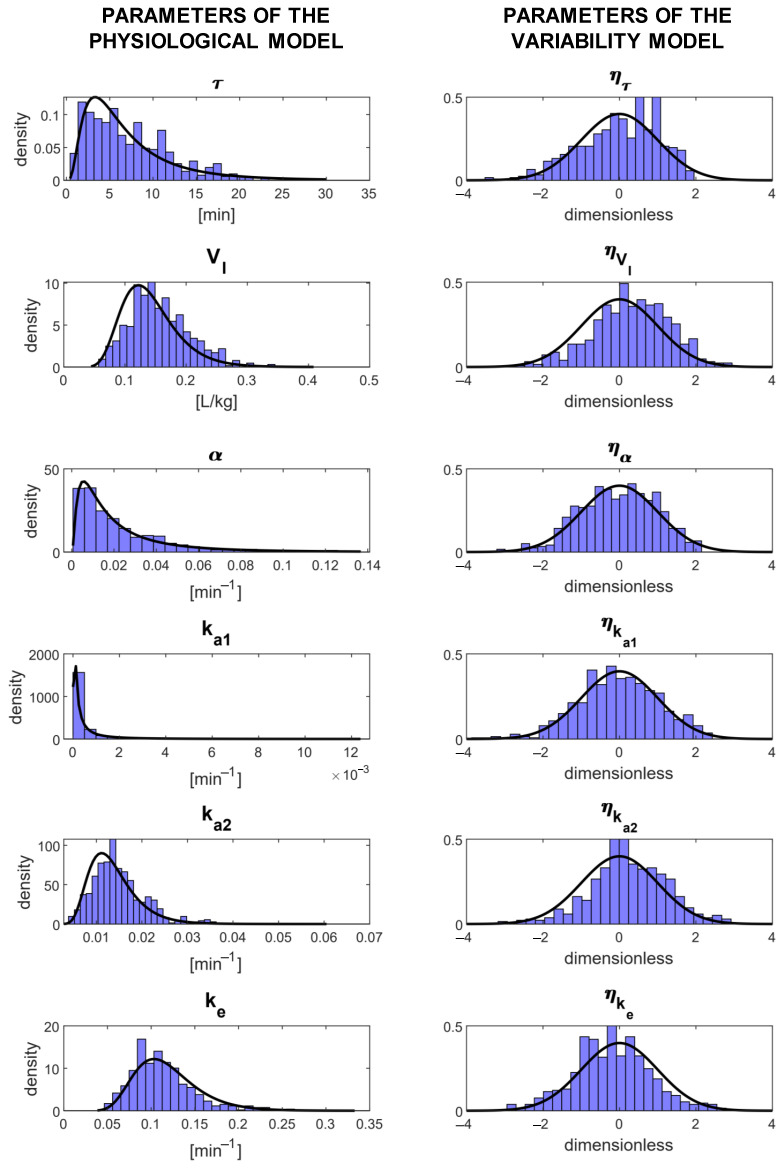
Parameter distributions of Model 11. Left panels: individual parameter distributions compared with the corresponding theoretical distributions (black line). Right panels: random effect distributions compared with a standard Gaussian (black line).

**Figure 6 metabolites-11-00235-f006:**
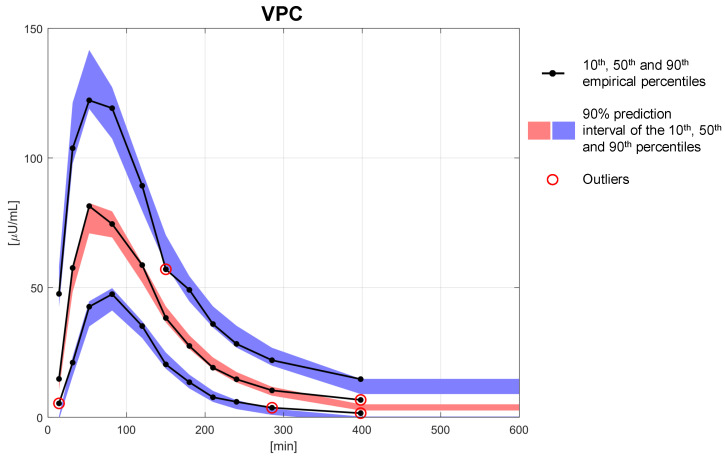
Visual predictive check obtained with the selected variability model (Model 11). The 90% prediction intervals of the 10th (blue lower area), 50th (red central area), and 90th (blue upper area) percentiles are compared with the 10th, 50th, and 90th empirical percentiles (black solid lines). Outliers are marked with red circles.

**Table 1 metabolites-11-00235-t001:** Summary of the tested  models of the parameter variability, ordered by ascending number of parameters. From left to right: model number; the additional parameter $\beta_{k,i}$ used to introduce the covariate $c_{k}$ in the variability model of $\psi_{i}$; the additional parameter $\rho_{i_{1},i_{2}}$ used to introduce a correlation between the random effects of the parameters $\psi_{i_{1}}$ and $\psi_{i_{2}}$; the estimate precision expressed as the mean RSE and the number of parameters showing RSE > 50%; percentages of the subjects that passed the Shapiro–Wilk (SW) test and the runs test; the value of the BICc. The selected model is highlighted in gray.

Model	Covariate	Correlation	Number of	Precision of the Estimates		% of Subjects that Passed:		BICc
Number	Coefficients	Parameters	Parameters	Mean RSE	RSE >50%	SW Test	Runs Test	
1	-	-	18	8.29	0	89.7	89.7	14,535.48
2	-	$\rho_{V_{I},k_{a2}}$	19	9.57	0	92.2	88.8	14,523.40
3	-	$\rho_{V_{I},k_{a2}}$ $\rho_{V_{I},k_{e}}$ $\rho_{k_{a2},k_{e}}$	21	9.35	0	90.5	88.8	14,494.36
4	$\beta_{\ln(AGE),\tau }$	$\rho_{V_{I},k_{a2}}$ $\rho_{V_{I},k_{e}}$ $\rho_{k_{a2},k_{e}}$	22	11.11	1	90.5	88.8	14,510.54
5	$\beta_{BMI,V_{I}}$	$\rho_{V_{I},k_{a2}}$ $\rho_{V_{I},k_{e}}$ $\rho_{k_{a2},k_{e}}$	22	32.24	1	91.4	88.8	14,506.42
6	$\beta_{BMI,k_{e}}$	$\rho_{V_{I},k_{a2}}$ $\rho_{V_{I},k_{e}}$ $\rho_{k_{a2},k_{e}}$	22	10.95	0	90.5	88.8	14,505.33
7	$\beta_{BSA,k_{a2}}$	$\rho_{V_{I},k_{a2}}$ $\rho_{V_{I},k_{e}}$ $\rho_{k_{a2},k_{e}}$	22	10.48	0	90.5	89.7	14,495.18
8	$\beta_{BMI,\tau }$	$\rho_{V_{I},k_{a2}}$ $\rho_{V_{I},k_{e}}$ $\rho_{k_{a2},k_{e}}$	22	11.61	1	90.5	88.8	14,487.14
9	$\beta_{BW,k_{a2}}$	$\rho_{V_{I},k_{a2}}$ $\rho_{V_{I},k_{e}}$ $\rho_{k_{a2},k_{e}}$	22	9.58	0	91.4	89.7	14,484.42
10	$\beta_{\ln\left(BW\right),k_{a2}}$	$\rho_{V_{I},k_{a2}}$ $\rho_{V_{I},k_{e}}$ $\rho_{k_{a2},k_{e}}$	22	8.59	0	91.4	88.8	14,477.76
11	$\beta_{BMI,k_{a2}}$	$\rho_{V_{I},k_{a2}}$ $\rho_{V_{I},k_{e}}$ $\rho_{k_{a2},k_{e}}$	22	9.11	0	91.4	88.8	14,476.86
12	$\beta_{BMI,\tau }$; $\beta_{BMI,k_{a2}}$	$\rho_{V_{I},k_{a2}}$ $\rho_{V_{I},k_{e}}$ $\rho_{k_{a2},k_{e}}$	23	11.01	1	91.4	87.9	14,481.72
13	$\beta_{BH,k_{a2}}$; $\beta_{BW,k_{a2}}$	$\rho_{V_{I},k_{a2}}$ $\rho_{V_{I},k_{e}}$ $\rho_{k_{a2},k_{e}}$	23	10.03	0	91.4	88.8	14,478.24
14	$\beta_{BMI,k_{a2}}$; $\beta_{\ln(AGE),\tau }$	$\rho_{V_{I},k_{a2}}$ $\rho_{V_{I},k_{e}}$ $\rho_{k_{a2},k_{e}}$	23	10.67	1	91.4	87.9	14,475.43

**Table 2 metabolites-11-00235-t002:** Estimated population parameters of the model and their corresponding RSE.

Estimates of the population parameters				
	Parameter	Estimated Value	Unit of Measurement	RSE
Fixed effects	$\tau^{pop}$	5.62	min	7.5
	$V_{I}^{pop}$	0.135	L/kg	3.58
	$\alpha^{pop}$	0.0155	min${}^{-1}$	10.5
	$k_{a1}^{pop}$	0.000134	min${}^{-1}$	36.9
	$k_{a2}^{pop}$	0.0128	min${}^{-1}$	3.43
	$k_{e}^{pop}$	0.113	min${}^{-1}$	3.33
	$\beta_{BMI,k_{a2}}$	−0.0865	m${}^{2}$/kg	13.6
Standard deviations of the random effects	$\omega_{\tau }$	0.731	dimensionless	8.81
	$\omega_{V_{I}}$	0.319	dimensionless	7.17
	$\omega_{\alpha }$	1.02	dimensionless	7.98
	$\omega_{k_{a1}}$	1.86	dimensionless	11.8
	$\omega_{k_{a2}}$	0.322	dimensionless	7.23
	$\omega_{k_{e}}$	0.303	dimensionless	6.51
Correlations between random effects	$\rho_{V_{I},k_{a2}}$	0.681	dimensionless	8.19
	$\rho_{V_{I},k_{e}}$	−0.579	dimensionless	11.1
	$\rho_{k_{a2},k_{e}}$	−0.506	dimensionless	14
Error model parameters	$a_{1}$	1.96	μU/mL	8.85
	$b_{1}$	0.0773	dimensionless	7.66
	$a_{2}$	2.33	μU/mL	5.47
	$b_{2}$	0.0516	dimensionless	5.79
	$a_{3}$	2.77	μU/mL	5.17
	$b_{3}$	0.0559	dimensionless	5.76

**Table 3 metabolites-11-00235-t003:** Summary of the individual parameter distributions obtained from model identification. From left to right: the minimum, the first quartile (25%), the median, the third quartile (75%), and the maximum value.

Estimates of the individual parameters						
Parameter	Min	Q1	Median	Q2	Max	Unit of Measurement
τ	0.748	4.03	5.98	10.5	21.9	min
$V_{I}$	0.0741	0.121	0.146	0.17	0.29	L/kg
α	0.00364	0.00968	0.0192	0.0334	0.116	min${}^{-1}$
$k_{a1}$	0.000079	0.000118	0.000132	0.000166	0.00824	min${}^{-1}$
$k_{a2}$	0.00418	0.0107	0.0136	0.0169	0.0313	min${}^{-1}$
$k_{e}$	0.046	0.0974	0.112	0.13	0.195	min${}^{-1}$

## Data Availability

Restrictions apply to the availability of these data. Data was obtained from Alan Krasner (Biodel Inc.), Olivier Soula (Adocia) and Gregory Meiffren (Adocia) and are available from the authors with the permission of Alan Krasner (Biodel Inc.), Olivier Soula (Adocia) and Gregory Meiffren (Adocia).

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
