# Peer review of "Modeling Between-Subject Variability in Subcutaneous Absorption of a Fast-Acting Insulin Analogue by a Nonlinear Mixed Effects Approach"

_metabolites, 2021, doi:10.3390/metabo11040235_

Round 1

Reviewer 1 Report

The present ms investigates the application of nonlinear mixed effects model (NLME) to insulin data. Overall, I positive about the ms and have a few remarks.
My points are as follows:
1.    For a relatively small number of subjects, Bayesian approaches could be advantageous for estimating the NLME. They should be mention in the introduction or the discussion section.
2.    P. 4, Equation 2: Should it read as d_j?
3.    Line 125: What is a ”\psi-dimensional function“?
4.    P. 5, Equation 3: I assume that the dots on the left-end side of the equation refer to derivatives with respect to t. The dot should be right above $I$ and bolder. Please look at, for example, https://tex.stackexchange.com/questions/44017/dot-notation-for-derivative-of-a-vector.
5.    Equation 3: If the \dot refers to a multiplication of real numbers in the equation, just omit it as usual in mathematical notation.
6.    Equation 4 and others. Please write ”exp“ and ”ln“ in the non-italic font.
7.    Equation 4: Is the \dot used here a scalar product? If yes, mention it for the first time when using the notation. If the \dot is just ordinary multiplication, omit the multiplication symbol.-
8.    Lines 138ff.: The remarks about missing data are a bit strange. Even in the case of missing at random data, just setting a missing value to a reference value of a covariate will introduce some bias.
9.    153: Should it read as ”\epsilon(t) \sim“?
10.    172ff.: A particular software is mentioned. Can the model also be estimated using standard software like R, SAS, or Stata?
11.    182ff.: Why is it important to assess the normality assumption of residuals (and random effects)? If the property seems relevant for valid statistical inference (i.e., standard errors), bootstrapping of subjects could be alternatively employed. If it is assessed because the meaning of model parameters would only be plausible, alternative distributional assumptions could be considered, such as Box-Cox or Yeo-Johnson distributions, a nonparametric NLME model, or mixture distributions. In any case, reporting the proportion of subjects that fail the Shapiro-Wilk test does not make sense at all, in my opinion. 
12.    182 ff.: Present a formula for RSE.
13.    184: ”N“ in italic.
14.    P. 9: Table 2 is poorly formatted. Consider merging information of Table 2 and Table 3 into one table.
15.    260ff.: Present size of correlations in addition to the p values that are merely a function of sample size.

Author Response

Thank you for your comments.

Please see the attachment whit our responses written in red, and the updated manuscript where the modifications were highlighted in red.

Reviewer 2 Report

A lot of work has been done in current study by authors and the results are really very interesting.

 I think that authors should add a paragraph about the usefulness of the proposed model in insulin pump devices and in algorithms those used for subcutaneous insulin doses in patients with type 1 Diabetes Mellitus. Only, one type of insulin analogues have been used.

I think that, the methodology is appropriate. The manuscript is generally clearly written and the discussion/ conclusion is acceptable.

   Overall, the data are of interest.

Author Response

Thank you very much for the encouraging comment.

We have added two sentences at page 2 and at page 15 reporting the usefulness of the proposed model for developing automated algorithms and infusion systems. Please see the attached file where we highlighted the updates in red.

Reviewer 3 Report

The article deals with the use of nonlinear mixed effects method for modeling of between subject variability. The issue of proper timing and dosing of the drug is important in terms of achieving optimal therapeutic effects. Insulin is one of the substances whose correct timing and dosing is still problematic, mainly due to the high variability in the absorption process, which can lead to dangerous fluctuations in the patient's blood glucose levels. Creating a model that is able to describe this variability will help in optimizing the optimal treatment setting for the patient. The article is a step towards the creation of such a tool and I believe that the authors will be able to develop the presented model to the set goals. I wish you much success. 

Author Response

Thank you very much for the encouraging comment.

In case you want to check the modifications that have been made, we are attaching the file where we highlighted the updates in red.

Round 2

Reviewer 1 Report

The revised ms and the response letter addressed my comments.
A few points are still left:
1.    P. 6: I agree with the authors that using a reference value for each person makes sense for interpretation. However, I still think that missing values are not automatically handled and will typically lead to biased estimates (at least in missing at random situations). 
2.    Authors should mention alternative but widespread software packages which allow the estimation of the models. In my opinion, alternative open-source software should be mentioned because it stands against the open science movement to rely on closed-source commercial software packages.
3.    I am still thinking that assessing the normality of residuals does not make sense. The normal distribution assumption of random effects is likely to be more critical. In the last review, I was asking for which reason normality should be assessed. The authors just stated that it is ”standard practice “ which is-of course-not a satisfactory response. If the ms were submitted in my field of substantive research, I would recommend rejecting the paper if the authors insist on including the normality tests still. The authors can decide how they want to proceed with my comment, but would urge them to think harder about what one is doing in statistical practice for whatever reason.
